# B-Cell Lymphoma of the Middle Ear Treated with Multidrug Chemotherapy in a Cat

**DOI:** 10.3390/vetsci10090585

**Published:** 2023-09-21

**Authors:** Tomoko Takahashi, Hiroyuki Nagata, Hirotaka Kondo

**Affiliations:** 1Department of Veterinary Medicine, College of Bioresource Sciences, Nihon University, 1866 Kameino, Fujisawa 2520880, Kanagawa, Japan; 2Noppo Animal Hospital, 2-12-2 Okamoto, Kamakura 2470072, Kanagawa, Japan

**Keywords:** facial nerve, hypoglossal nerve, lymphoma, nasopharyngeal polyp, otitis media, prednisolone

## Abstract

**Simple Summary:**

Feline lymphoma can arise in a variety of sites, with the pathogenesis differing according to the site of origin. Although cases of primary lymphoma of the middle ear have been reported, information on treatment efficacy and prognosis remains lacking. In this study, an 11-year-old, spayed female mixed-breed cat was diagnosed with primary lymphoma of the middle ear. Prednisolone had been used as an anti-inflammatory drug with suspicion of otitis media for 4 months before diagnosis of lymphoma. We suspected that the cat could have maintained in good general condition because prednisolone was one of the anticancer agents for lymphoma. Multidrug anticancer chemotherapy was started and proved to be effective. The patient’s condition had been stable for another 4 months. Unfortunately, invasion from the middle ear to the nasopharynx caused difficulty with nasal breathing, liver involvement was suspected, and she died of seizures on day 228. Primary feline lymphoma of the middle ear might respond well to chemotherapeutic treatment. The prognosis for this type of lymphoma may not be as terrible as previous reports.

**Abstract:**

Primary lymphoma of the middle ear is rare in cats, and little information has been accumulated on the treatment and course. An 11-year-old spayed female mixed-breed cat was brought to our hospital with facial nerve paralysis, which had been stable using prednisolone for 3 months. She was diagnosed with B-cell lymphoma of the right middle ear after otoscopic sampling, which showed evidence of the filling of bilateral tympanic bullae on computed tomography. Hepatic involvement was suspected, and she was treated with multidrug chemotherapy, resulting in clinical remission. On day 176, magnetic resonance imaging and computed tomography were performed at checkup, revealing tumor invasion into the nasopharyngeal region and the recurrence of hepatic lesions without any brain abnormality. Nasal congestion then worsened, and the patient died on day 228 after presenting with generalized seizures. Clinicians should be mindful of middle ear lymphoma as a differential diagnosis in cats who present with a sign of otitis media, especially whose condition does respond to corticosteroid treatment. The prognosis for feline middle ear lymphoma might not be as poor as previous reports, and multidrug chemotherapy might result in remission of the disease.

## 1. Introduction

Feline extranodal lymphoma occurs in a variety of sites and often shows characteristic behaviors at different sites. For example, nasal lymphoma is associated with longer survival times, and lymphoma arising in the central nervous system shows shorter survival [1,2]. On the other hand, lymphoma of the middle ear in cats is very rare, and the prognosis has been reported to be poor: a case of T-cell lymphoma diagnosed by tympanostomy was euthanized before treatment was started [3]; a case of non-B and non-T cell lymphoma was euthanized because of brain infiltration and apnea that developed after anesthesia induction [4]; a case of stage 1 T-cell lymphoma that was feline leukemia virus (FeLV)-positive died suddenly of seizures 7 days after completion of radiotherapy for middle ear and brain infiltration, with invasion to the liver and spleen on autopsy [5]. All patients in these previously reported cases died within a short period. As a result, little is known about the pathogenesis of this type of lymphoma.

In this study, we describe the clinical presentation and outcome of a cat with primary lymphoma of the middle ear, treated with a multidrug chemotherapy protocol. Follow-up was conducted over a prolonged period.

## 2. Case Presentation

An 11-year-old spayed female mixed-breed cat was referred to our hospital with a chief complaint of chronic facial nerve paralysis. She had presented with sudden nystagmus and staggering gait 3 months earlier. Magnetic resonance imaging (MRI) was performed at another hospital, revealing suspected inflammatory granulation in the right tympanic bulla. Otitis media was tentatively diagnosed, and the patient was started on prednisolone at 5 mg/cat/day with no recurrence of nystagmus. However, the dose could not be reduced because withdrawal from prednisolone resulted in decreased vigor and appetite. The patient had displayed transient right torsional head tilt and narrowing of the right eyelid fissure 2 weeks earlier.

At the time of initial examination, she weighed 3.60 kg (body condition score 3/5) with no obvious abnormalities in posture or gait. Minimal cerumen was present, and cervical lymph nodes were not enlarged. Cranial neurologic examination revealed disturbance of the right facial nerve and abnormalities of the hypoglossal nerve (narrowing of the right eyelid fissure, loss of right menace reaction, loss of right eyelid reflex, decreased right corneal reflex, decreased right maxillary and mandible perception, and folded tongue). No history of seizures was elicited. Complete blood count, blood biochemistry, and coagulation test were within the reference ranges, and urinalysis showed no obvious abnormalities. Negative results were obtained for feline immunodeficiency virus antibody and FeLV antigen test (SNAP FIV/FeLV Combo Test; IDEXX Laboratories, Westbrook, ME, USA). Computed tomography (CT) of the head without anesthesia or contrast media was performed to confirm the status of the middle ear, revealing space-occupying lesions in bilateral tympanic bullae (Figure 1). No bony changes or other obvious abnormalities were identified outside the middle ear, including in the nasal cavity. The lesions within the tympanic bullae resembled effusion or soft tissue, so infection and neoplastic disease were the differential diagnoses.

On day 28, the weight of the patient had decreased to 3.15 kg (body condition score 2/5), although vigor and appetite remained otherwise stable. The owner reported frequent spills of food, probably due to abnormality of the hypoglossal nerve. After inducing general anesthesia, the middle ear was examined by video-otoscope. The right tympanic membrane was filled with a cauliflower-shaped mass, precluding observation of the entire tympanic bulla. A portion of the mass was sampled using endoscopic forceps. Observation of the left ear canal revealed a thickened tympanic membrane, the surface of which was sampled. Histopathological and immunohistochemical examination diagnosed the mass in the right tympanic cavity as poorly differentiated large B-cell lymphoma (Figure 2). No obvious abnormalities were seen for the sample from the left tympanic membrane.

On day 42, whole-body CT without anesthesia or contrast media was performed for clinical staging. The middle ears appeared unchanged, but diffuse changes were observed in the liver (Figure 3A). Cytology of the liver lesions was not performed per the owner’s request. The tentative diagnosis was middle ear lymphoma with hepatic involvement, representing clinical stage 4 disease. After discussing the treatment plan with the owner, subcutaneous L-asparaginase was administered at 400 IU/kg (Kyowa Kirin, Tokyo, Japan). Prednisolone was continued at 5 mg/cat/day.

On day 56, right eyelid opening had improved. Follow-up whole-body CT without anesthesia or contrast media showed that the dorsolateral lesion of the right tympanic bulla had disappeared, the hepatic lesions were obscured (Figure 3B), and liver swelling had improved. L-asparaginase was evaluated as effective, and a second dose was administered.

On day 68, multidrug chemotherapy was started using the modified CHOP protocol [6]; the patient was treated with vinblastine at 1.5 mg/m^2^ intravenously (Nippon Kayaku Co., Tokyo, Japan), cyclophosphamide at 10 mg/kg intravenously (Shionogi Co., Osaka, Japan), and doxorubicin at 1 mg/kg intravenously (Sandoz K-K., Tokyo, Japan). No significant adverse effects were seen in terms of weight change or results of blood tests, and the protocol was generally administered as planned, with three postponements of administration due to deterioration in the physical condition of the patient, including decreased appetite and dizziness. Considering the boisterous character of the case, the patient was sedated with butorphanol at 0.4 mg/kg intramuscularly (Meiji Animal Health Co., Tokyo, Japan) each time catheter needle was placed, and the anticancer drug administration was completed without any problems. Neurological abnormalities were not completely resolved, but since no increases in liver enzyme levels were noted, the patient was considered to have achieved clinical remission, and the protocol was continued every other week after day 152. Prednisolone administration at 5 mg/cat/day was continued because the patient’s condition worsened when the dose was reduced.

On day 176, the patient developed an unstable appetite and nasal obstruction sounds. Intraoral examination revealed swelling of the soft palate. Blood tests revealed no particular abnormalities. Contrast-enhanced whole-body CT under anesthesia showed unchanged filling of bilateral tympanic bullae and no obvious bony abnormalities. A mass lesion with heterogeneous contrast enhancement was found on the ventral side of the right tympanic bulla, almost completely obstructing the nasopharyngeal passage (Figure 4A). The right medial retropharyngeal lymph node was contiguous with the ventral lesion of the tympanic bulla and showed heterogeneous contrast enhancement. Multiple contrast-enhanced mass lesions were noted in the liver (maximum diameter, 18 mm) (Figure 4C,D). No obvious intrathoracic or intra-abdominal organ abnormalities were apparent other than in the liver. MRI of the head revealed no obvious meningeal lesions and other abnormalities of the brain (Figure 4B). The tentative diagnosis was spread of lymphoma in the right middle ear to the Eustachian tube and formation of a mass in the nasopharyngeal region.

L-asparaginase was administered as rescue therapy on day 180, but the patient progressively lost appetite due to nasal obstruction, so 10 mg of lomustine (51 mg/m^2^) (Bristol-Myers Squibb, Saint-Laurent, QC, Canada) was administered orally on day 187 [7]. Nasal obstruction subsequently improved, and appetite was restored, but pupillary asymmetry appeared. L-asparaginase was again administered on day 201, but pupillary asymmetry did not improve, and the patient continued to have unstable appetite, so supportive care was provided. As her condition remained relatively stable, L-asparaginase was administered on day 216 and cyclophosphamide on day 222 while continuing supportive care. The patient developed recurrent nystagmus and generalized seizures on day 226 and was treated symptomatically but died on day 228. No postmortem examination was performed.

## 3. Discussion

Prednisolone alone appeared effective in inducing partial tumor response, and the cat subsequently demonstrated an excellent response to the addition of a multidrug chemotherapy protocol, remaining in clinical remission for several months. Feline middle ear lymphoma is rare, and the patients in all previously reported cases have died within a short period, so the pathogenesis remains unclear [3,4,5]. Whether the course of our case was typical is not known, so more cases should be accumulated in the future.

The patient was very stable on 5 mg/cat/day of prednisolone (equivalent to 1.5–2.2 mg/kg/day), worsened when the dose was reduced, and recovered when the dose was restored. In feline extranodal lymphoma, treatment with prednisolone alone has been reported as effective in 7 of 10 cats [2]. Prednisolone may also be effective against lymphoma of the middle ear. On the other hand, because of this high efficacy, the use of prednisolone as a symptomatic treatment before diagnosis may delay lymphoma diagnosis. If clinical signs related to the middle ear improve with prednisolone, lymphoma may need to be included among the differential diagnoses, and further examinations should be suggested to owners.

The patient in this case was evaluated as having responded well to anticancer treatment, as remained clinically stable for 4 months after initiating multidrug chemotherapy. However, the patient had been on prednisolone for 4 months prior to the initiation of combination chemotherapy, which may have induced multidrug resistance, resulting in reduced effectiveness of the anticancer drugs. Among feline patients with extranodal lymphoma treated with multidrug chemotherapy and achieving complete remission, those cases in which prednisolone had been used before starting chemotherapy were reported to have significantly shorter survival [2]. In addition, a recent study of feline lymphoma cells has suggested that the efficacy of doxorubicin and vincristine is reduced, i.e., multidrug resistance may be induced after the use of prednisolone [8]. On the other hand, pretreatment with corticosteroids did not affect progression-free survival of laryngeal or tracheal lymphoma [9]. In our case, it was unclear whether the pretreatment with prednisolone had altered the response to multidrug chemotherapy. Whichever it may be, the fact that this case responded well to chemotherapy, even after prolonged use of prednisolone, is favorable information in the treatment of this type of lymphoma, whose diagnosis and adequate treatment initiation were significantly delayed. In previous case reports, brain invasion was observed in the early stages of the disease [4,5], and brain infiltration was finally suspected in this case as well. Lomustine showed short-term efficacy as a rescue agent in our patient. The early use of anticancer agents that penetrate the blood–brain barrier, such as lomustine, might be useful for this type of lymphoma.

Multidrug chemotherapy was selected in our patient because hepatic involvement was suspected at the time treatment was started. If the diagnosis had been made at clinical stage 1, in a localized state, radiotherapy might have been a good option. In cases of intranasal lymphoma, which has a relatively high incidence as extranodal lymphoma and in the absence of other organ involvement, long-term remission can be achieved with radiotherapy of the nasal lesions alone [10]. In human medicine, radiotherapy and/or chemotherapy were reported to be effective for ear lymphoma [11]. Effective treatment for feline middle ear lymphoma has yet to be defined.

The present case suggests that the spread of primary lymphoma of the middle ear might show a predilection for the liver. Although we did not perform a cytologic examination of the liver because the owner declined to consent, the hepatic lesions, in this case, were considered likely to represent dissemination from the lymphoma since the diffuse changes in the liver temporarily disappeared after administration of L-asparaginase. Hepatic involvement was also suspected on day 176, although hepatic enzymes were not elevated during follow-up. Santagostino et al. also found hepatic involvement at autopsy [5], and hepatic dissemination might be common for lymphoma of the middle ear. In feline extranodal lymphoma, follow-up should be based on knowledge of the pathophysiologic characteristics associated with each primary site.

Feline primary middle ear lymphoma can present a diagnostic challenge. It took a relatively long time to provide a definitive diagnosis in this case as well as in a previously published [5], suggesting that clinical signs might mimic others of non-neoplastic diseases, such as otitis media and nasopharyngeal polyps. Moreover, there seems to exist no image findings typical of middle ear lymphoma. Given the common finding of middle ear effusion on CT [12], undiagnosed middle ear lymphoma might be underestimated. Biopsy of the middle ear is relatively challenging because it usually requires general anesthesia, otoscopy, and, ideally, a CT scan. Nevertheless, a biopsy should be suggested to owners of cats presenting with prednisolone-responsive middle ear-related symptoms, including facial paralysis. In a report on ear lymphoma in human patients, authors recommended excluding a malignancy when a patient showed therapy-resistant otitis and/or peripheral facial paralysis [11].

There were several limitations in this study. First, after the diagnosis of lymphoma, it took almost a month before the patient was started on modified CHOP chemotherapy. This was due to hesitation to start treatment because the incidence of this type of lymphoma was low, and the outcome of treatment was unknown. After a trial administration of L-asparaginase proved to be effective, the decision was made to start a multidrug regimen. Second, several CT scans were performed without anesthesia or contrast media to reduce patient burden; only abnormalities with variable attenuation changes were detected. It is possible that more abnormalities could have been detected if contrast media had been administered and initial staging could have been changed. Furthermore, no cytological specimen was obtained from the liver, and hepatic involvement was only suspected based on the images. Whether brain invasion occurred antemortem had also not been confirmed because an autopsy was not performed.

## 4. Conclusions

Clinicians should be mindful of middle ear lymphoma as a differential diagnosis in cats who present with a sign of otitis media, especially whose condition does respond to corticosteroid treatment. The prognosis for this type of lymphoma may not be as poor as previous reports. Even if the diagnosis is delayed, multidrug chemotherapy may result in remission of the disease. Further accumulation of patients with middle ear lymphoma is needed to characterize this type of extranodal lymphoma.

## Figures and Tables

**Figure 1 vetsci-10-00585-f001:**
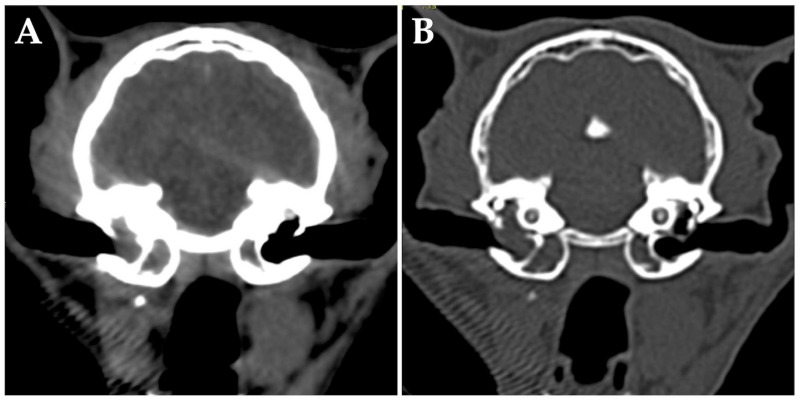
Transverse computed tomography (without anesthesia or contrast media) of the head at the level of the middle ear of the cat at first presentation in the soft tissue window (**A**) and bony window (**B**). The ventromedial compartment of the left tympanic bulla and the ventromedial and dorsolateral compartments of the right tympanic bulla are occupied by fluid/soft tissue attenuating materials. No bony abnormalities are evident.

**Figure 2 vetsci-10-00585-f002:**
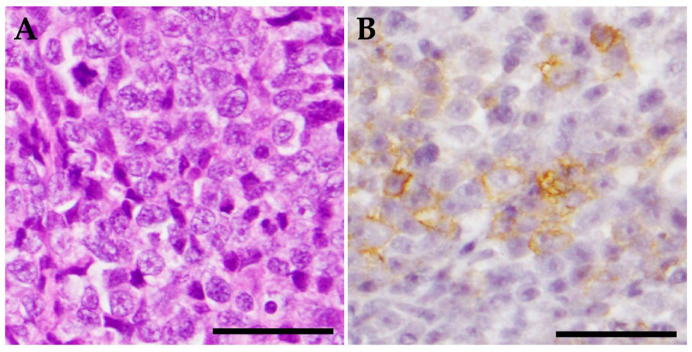
Histopathological and immunohistochemical examination. The tissue sample from the right tympanic bulla comprises tumor tissue. Tumor cells show moderate anisocytosis and anisokaryosis, with 6 mitotic figures per 10 high-power fields. This represents poorly differentiated large-cell lymphoma. (**A**) Hematoxylin and eosin stain, ×400. Scale bar = 20 μm. (**B**) Neoplastic cells are positive for anti-CD20 antibody (anti-CD20 rabbit polyclonal antibody, diluted 1:800; Funakoshi, Tokyo, Japan). Mayer hematoxylin counterstain, ×400. Scale bar = 20 μm.

**Figure 3 vetsci-10-00585-f003:**
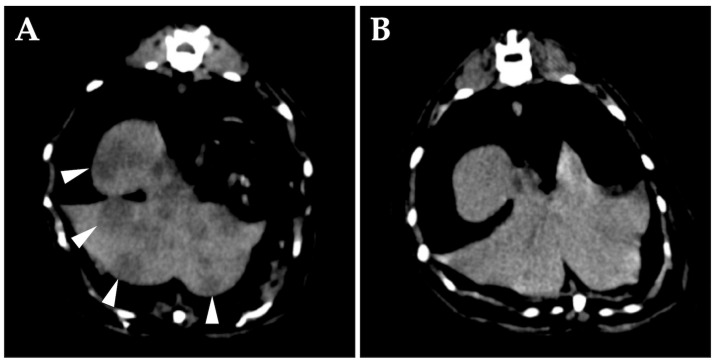
Transverse computed tomography (CT) (without anesthesia or contrast media) of the abdomen at the level of the liver on day 42. (**A**) Numerous hypo-attenuating nodular lesions are seen in all lobes of the liver (arrowheads). (**B**) CT image of the same position on day 56. The hepatic lesions are obscured.

**Figure 4 vetsci-10-00585-f004:**
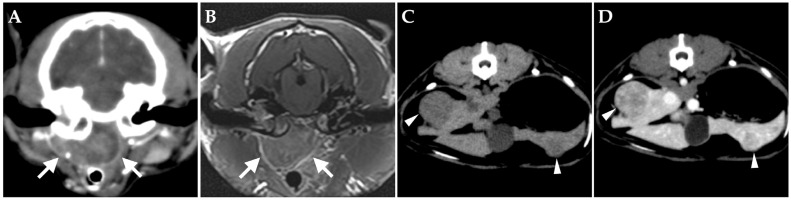
Transverse images on day 176. (**A**,**B**) Computed tomography (CT) (**A**) and magnetic resonance imaging (**B**) of the head at the level of the middle ear. A mass lesion showing heterogeneous enhancement from contrast media is present in the nasopharyngeal region (arrow). (**C**,**D**) On CT of the abdomen at the level of the liver, multiple hypo-attenuating, contrast-enhanced nodules are present in the liver (arrowheads) ((**C**) pre-contrast; (**D**) post-contrast).

## Data Availability

The data presented in this study are available in the manuscript.

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
