# Peer review of "B-Cell Lymphoma of the Middle Ear Treated with Multidrug Chemotherapy in a Cat"

_vetsci, 2023, doi:10.3390/vetsci10090585_

Round 1

Reviewer 1 Report

this is a well written case report of an unusual lymphoma in the middle ear with hepatic mets. in a cat. Unfortunately no post mortem was obtain or at least was not reported to corroborate the diagnosis and spread of the disease post chemotherapy.  Therefore it is not shown that it did or did not spread to the brain despite initial improvement on the chemotherapy followed by seizures which might suggest that it did.  Some comment on this consideration might be added to the discussion,  as well as why the chemotherapy was changed on  day 68.    

the English is good with only minor changes corrections needed. 

Author Response

Thank you for your remarks. I specified as a limitation that liver involvement was not pathologically confirmed and brain invasion was also not confirmed at autopsy.

“Second, several CT scans were performed without anesthesia or contrast media to reduce patient burden; only abnormalities with variable attenuation changes were detected. It is possible that more abnormalities could have been detected if contrast media had been administered, and initial staging could have been changed. Furthermore, no cytological specimen was obtained from the liver, and hepatic involvement was only suspected based on the images. Whether brain invasion occurred antemortem had also not been confirmed because an autopsy was not performed. “

The reason for changing the protocol on day 68 was as follows: 

Since there were only poor prognostic reports, we were unsure for a while whether to start anticancer therapy after the diagnosis of lymphoma. The first L-ASP was given as a trial, because it could be administered without the sedation and catheter placement. The second L-ASP was administered because the first one was effective without side effects, and the owner was still unsure whether to start multidrug anticancer therapy. Eventually, on day 68, the owner decided to start the multidrug anticancer therapy, so we started it.

In fact, it took a long time to diagnose the disease, and even longer to begin treatment once it was diagnosed. All because the incidence of this type of lymphoma was low, and the outcome of treatment was unknown.

We have added these hesitations to the discussion.

“First, after the diagnosis of lymphoma, it took almost a month before the patient was started on modified CHOP chemotherapy. This was due to hesitation to start treatment, because the incidence of this type of lymphoma was low, and the outcome of treatment was unknown. After a trial administration of L-asparaginase proved to be effective, the decision was made to start a multidrug regimen. “

Reviewer 2 Report

The study provide information aboutthe treatment efficacy and prognosis of a rare case of primary lymphoma in the middle ear of a domestic cat, which has limited existing information. The study presents a comprehensive case report, detailing the clinical history, diagnosis, treatment approach, and eventual outcomes of the cat's condition. The manuscript highlights the effectiveness of prednisolone and multidrug chemotherapy, as well as the challenges posed by metastasis and complications.

The study contributes to the understanding of a rare condition in feline medicine. The detailed clinical presentation and treatment course provide useful information for both veterinarians and researchers in the field. The discussion of treatment efficacy and the challenges encountered adds depth to the case study. The manuscript is generally well-organized and informative, but there are a few areas that could benefit from improvement before publication.

1. Abstract: The abstract provides a simple summary of the case, but it would benefit from incorporating some of the key insights from the discussion section.

2. Implications for Clinical Practice: In the discussion section, consider expanding on the practical implications of the findings for clinical practitioners. How can the insights gained from this case study guide diagnostic and treatment decisions in similar cases?

Author Response

Thank you very much for your useful suggestions. There were many parts, including limitation, where explanations were insufficient. By rewriting the paper according to your suggestions, we were able to make it more reader-friendly and scholarly.

1. The significance of this case report has been added to the abstract:

Clinicians should be mindful of middle ear lymphoma as a differential diagnosis in cats who present with a sign of otitis media, especially whose condition does respond with corticosteroid treatment. The prognosis for feline middle ear lymphoma might not be as bad as previous reports, and multidrug chemotherapy might result in remission of the disease. 

2. As you suggested, we have added a couple of suggestions that would be helpful to clinicians: 

“In our case, it was unclear whether the pretreatment with prednisolone had altered the response to multidrug chemotherapy. Whichever it may be, the fact that this case responded well to chemotherapy, even after prolonged use of prednisolone, is favorable information in the treatment of this type of lymphoma, whose diagnosis and treatment start are easily delayed. 

“Biopsy of the middle ear is relatively challenging, because it usually requires general anesthesia, otoscopy, and ideally CT scan. Nevertheless, biopsy should be suggested to owners of cats presenting with prednisolone-responsive middle ear-related symptoms, including facial paralysis. In a report of ear lymphoma of human patients, authors recommended to exclude a malignancy when patient showed a therapy-resistant otitis and/or a peripheral facial paralysis [11].